# Exploring the Impact of Antibiotics on Fever Recovery Time and Hospital Stays in Children with Viral Infections: Insights from Advanced Data Analysis

**DOI:** 10.3390/antibiotics13060518

**Published:** 2024-06-01

**Authors:** Mohammed Al Qahtani, Saleh Fahad AlFulayyih, Sarah Saleh Al Baridi, Sara Amer Alomar, Ahmed Nawfal Alshammari, Reem Jassim Albuaijan, Mohammed Shahab Uddin

**Affiliations:** Department of Pediatric, Ministry of National Guard Health Affairs, Dammam 31412, Saudi Arabia; alqahtanim17@mngha.med.sa (M.A.Q.); alfulayyihs@mngha.med.sa (S.F.A.); albaridis@mngha.med.sa (S.S.A.B.); alomarsr@mngha.med.sa (S.A.A.); alshammarian@mngha.med.sa (A.N.A.); albuaijanr@ngha.med.sa (R.J.A.)

**Keywords:** antibiotics, respiratory tract infections, pediatrics, hospitalization, fever, anti-bacterial agents, polymerase chain reaction

## Abstract

**Background:** Antibiotic overuse in pediatric patients with upper respiratory tract infections (UR-TIs) raises concerns about antimicrobial resistance. This study examines the impact of antibiotics on hospital stay duration and fever resolution in pediatric patients diagnosed with viral infections via a multiplex polymerase chain reaction (PCR) respiratory panel. **Methods:** In the pediatric ward of Imam Abdulrahman Bin Faisal Hospital, a retrospective cohort analysis was conducted on pediatric patients with viral infections confirmed by nasopharyngeal aspirates from October 2016 to December 2021. Cohorts receiving antibiotics versus those not receiving them were balanced using the gradient boosting machine (GBM) technique for propensity score matching. **Results:** Among 238 patients, human rhinovirus/enterovirus (HRV/EV) was most common (44.5%), followed by respiratory syncytial virus (RSV) (18.1%). Co-infections occurred in 8.4% of cases. Antibiotic administration increased hospital length of stay (LOS) by an average of 2.19 days (*p*-value: 0.00). Diarrhea reduced LOS by 2.26 days, and higher albumin levels reduced LOS by 0.40 days. Fever and CRP levels had no significant effect on LOS. Time to recovery from fever showed no significant difference between antibiotic-free (Abx0) and antibiotic-received (Abx1) groups (*p*-value: 0.391), with a hazard ratio of 0.84 (CI: 0.57–1.2). **Conclusions:** Antibiotics did not expedite recovery but were associated with longer hospital stays in pediatric patients with acute viral respiratory infections. Clinicians should exercise caution in prescribing antibiotics to pediatric patients with confirmed viral infections, especially when non-critical.

## 1. Introduction

The global health challenge today is significantly impacted by the growing problem of antibiotic resistance [1]. Initially celebrated for their effectiveness in fighting bacterial infections, antibiotics are now becoming less effective due to overuse and misuse [2,3]. The World Health Organization (WHO) has identified antibiotic resistance as a critical threat to global health, affecting food security and development [4]. This crisis is driven by the unnecessary use of antibiotics, their availability without prescription, and their use in agriculture, which is not related to human health [5,6]. Additionally, the development of new antibiotics is lagging, mainly due to economic challenges within the pharmaceutical industry [7]. This problem is particularly serious for children under five, who are very vulnerable to respiratory infections [8]. Due to antibiotic resistance, these infections can become more severe and prolonged, leading to longer hospital stays and increased death rates among these young patients (Laxminarayan et al. [9]).

The widespread misuse of antibiotics is a global issue, as highlighted by several studies. In the United States and the United Kingdom, there are consistent patterns of overprescribing antibiotics in pediatric settings, often for conditions caused by viruses [10,11,12,13]. This trend is also observed in European countries, despite their well-established health systems [14,15,16]. In Australia, concerns about antibiotic stewardship are growing, with estimates suggesting that nearly half of all antibiotic prescriptions may be unnecessary [17,18,19]. In India, the challenges are worsened by over-the-counter sales of antibiotics and limited diagnostic facilities [20,21]. Many other countries face significant hurdles due to the absence of structured antibiotic management programs and sufficient regulatory oversight [22,23,24]. The ineffectiveness of antibiotics against viral infections, especially acute viral respiratory tract infections in children, is well-documented [25,26,27]. Since antibiotics do not have antiviral properties, their use in such cases is not only ineffective but can also disrupt natural microbiota and lead to adverse antibiotic-related events [28,29].

In the context of escalating antibiotic resistance and clear evidence of the ineffectiveness of antibiotics for viral infections, there is a notable gap in knowledge [30,31]. Most of the studies were observational and reported associations linked rather than causality. Some studies, such as studies from Lee and Hassan (2019), Blatt et al. (2017), and Yen et al. (2019), show that antibiotics do not improve clinical outcomes or shorten hospital stays for children with viral respiratory tract infections (ARTIs). These studies call for better diagnostic tools to determine if antibiotics are necessary [32,33,34]. The findings of Fahey and Stocks (1998) and Petersen (2007), who brought attention to the overestimation of antibiotic advantages in pediatric viral illnesses [35,36], corroborate the limited usefulness of these medications in changing hospitalization lengths and fever recovery. Moreover, there are major gaps in the literature connecting antibiotic use to shorter hospital stays which has led Mustafa and Salman (2020) and Little and Rumsby (2005) to express concerns regarding the appropriate use of antibiotics [37,38]. In a similar vein, research by Pichichero et al. (2000) and Jain et al. (2001), highlights the fact that most children can recover from viral RTIs without antibiotics. These studies call for cautious prescribing to prevent needless problems and resistance [39,40]. The necessity for integrated diagnostic and treatment strategies is further demonstrated by the fact that rapid diagnostics, like the PCR multiplex respiratory panel, which was emphasized in studies by Kim et al. (2021) and Papan et al. (2020), improve diagnosis speed but do not significantly decrease length of stay or antibiotic use [41,42].

Despite clear evidence of their limited effectiveness, antibiotics are frequently prescribed for pediatric viral upper respiratory tract infections (URTIs), often based on misconceptions about their benefits. Research indicates that antibiotics do not significantly reduce fever duration or enhance clinical outcomes in viral URTIs [43,44,45,46]. Such misuse is exacerbated by widespread false beliefs regarding antibiotic efficacy for symptoms like fever, which prompt an inappropriate prescription rate of 40% to 75% in North America, often when fever exceeds 38.5 °C [47,48]. In Saudi Arabia, 27% of parents seek antibiotics for their children’s fever, illustrating a global issue [23]. The persistence of incorrect beliefs among parents about the role of antibiotics in treating fever and ear pain necessitates targeted educational interventions to correct these views and promote effective fever management strategies [49]. Antibiotics are reserved for specific instances, like influenza, to prevent secondary bacterial infections, but do not change the recovery timeline from the virus itself [50]. A meta-analysis by Fahey and Stocks showed that antibiotics do not improve outcomes in children with URTIs, with no significant reduction in symptom duration or hospitalization [35]. Petersen (2003) found that the number needed to treat with antibiotics to prevent one serious complication from an acute viral infection exceeds 4000, indicating their minimal role in such cases.

In fact, there is a lack of comprehensive research examining the impact of antimicrobial agents on key health outcomes such as hospital stay duration and fever resolution time in children with viral respiratory tract infections. This gap highlights the urgent importance of our current study. Our working hypothesis is that antibiotics do not significantly alter these outcomes and instead contribute to global antibiotic abuse and resistance. To evaluate these results, we used a survey-weighted linear regression model and the gradient boosting machine (GBM) method for propensity score matching [51,52]. These advanced data analysis methods allow us to compare cohorts treated with antibiotics against those not treated, focusing on the duration required for fever resolution and hospital stay lengths. While antibiotics are often criticized for their ineffectiveness against viral pathogens, they are crucial in preventing bacterial complications arising from influenza. With precise diagnostic practices that accurately identify bacterial co-infections, antibiotics can be prescribed more judiciously, ensuring their use is both necessary and appropriate.

## 2. Materials and Methods

This study employed a retrospective analysis of electronic medical records (EMRs) for pediatric patients admitted between October 2016 and December 2020 to the Pediatric ward at Imam Abdulrahman Bin Faisal Hospital in Dammam, Saudi Arabia. The proposal was approved by the Institutional Review Board (IRB), King Abdullah International Medical Research Center, Riyadh, Saudi Arabia, under the reference number RD20/004/D. The Hospital affiliated with the National Guard Health Affairs, has an overall capacity of approximately 100 beds. The pediatric ward has a designated capacity of 20 beds, with an average annual admission rate of 1200. Notably, during winter, admission rates frequently exceed the bed capacity of the ward.

For the purposes of our research, the age range of the pediatric population under investigation spanned from 1 month to 14 years. To be included in the cohort, children had to exhibit acute-onset respiratory symptoms and be diagnosed with acute viral respiratory tract infection (VRTI) through PCR multiplex respiratory panels conducted on nasopharyngeal aspirates. The key exclusion criteria were as follows: (1) Prematurity with a corrected age under 40 weeks; (2) Immunocompromised status; (3) Documented bacterial etiology evident from blood cultures, urine cultures, or respiratory panels; (4) Suspected sepsis with hemodynamic instability. (5) SARS-CoV-2 was excluded from the analysis.

Our research aimed to utilize gradient boosting machine algorithms to investigate the causal impact of antimicrobial agents on the recovery times from fever and the duration of hospital stay in non-critical pediatric patients diagnosed with acute VRTI.

### Co-Variates

In our analytical framework, we meticulously assimilated 29 confounders spanning demographic, clinical, and laboratory dimensions. Demographically, age and sex were cornerstone parameters. From a clinical perspective, our analysis delved deep into the nuances of disease duration, antecedent fever episodes, zenith-recorded temperatures, and trajectory to fever resolution. It is important to highlight that we operationalized ‘fever recovery’ as a phenomenon in which the fever abates within 72 h of its inaugural documentation, treating any protracted or non-diminishing febrile episode during the ensuing observation period as a censored phenomenon. In addition, our clinical lens also captured manifestations, such as cough, dyspnea, wheezing, diarrhea, and vomiting, coupled with the presence of any concurrent comorbid pathologies or coinfections. Shifting the gaze to the laboratory dimension, our investigatory matrix was bolstered by parameters including, but not limited to, hemoglobin concentrations, leukocyte metrics, neutrophil and lymphocyte enumerations, thrombocyte counts, C-reactive protein indices, creatinine benchmarks, alanine aminotransferase metrics, albumin gradients, respiratory intervention necessities, and the cumulative duration of hospital-based care. Collectively, these confounders meticulously architect a robust analytical scaffold, empowering our gradient boosting machine algorithms to dissect the potential ramifications of antimicrobial therapeutics on fever trajectories and hospitalization epochs in non-critical pediatric cohorts with acute viral respiratory tract pathologies.

Acute Viral Respiratory Tract Infection: An acute viral respiratory tract infection is a rapid-onset infection caused by one of several respiratory viruses, including influenza, respiratory syncytial virus (RSV), rhinovirus, coronavirus, adenovirus, and others. The infection typically affects the nose, throat, airways, or lungs. Clinically, ARTIs present with a constellation of symptoms including, but not limited to, cough, sore throat, nasal congestion, rhinorrhea, fever, malaise, and dyspnea. These symptoms can vary in intensity and may be accompanied by systemic features such as fever and body aches. Diagnosis is primarily clinical but can be confirmed through molecular testing, antigen detection, or viral culture. The management of ARTIs is generally supportive, with antiviral medications used for specific viruses and in particular patient populations [32].

Laboratory diagnosis of acute viral respiratory tract infection: A multiplex polymerase chain reaction (PCR) assay was employed to discern the genomic material of an extensive panel of 22 respiratory pathogens, including the recently added SARS-CoV-2. This multiplex real-time PCR analysis was meticulously executed with strict adherence to the protocols delineated by its manufacturer, See-gene, a renowned entity headquartered in Korea. The diagnostic tool of choice was the Any-plex™ II RV16 detection kit (Seegene Inc., Seoul, South Korea), an innovative product from Seegene’s repertoire tailored to identify a spectrum of viral pathogens. The detection capabilities of this kit encompass a range of viruses, notably adenovirus, influenza A and B, parainfluenza viruses (types 1 through 4), rhinovirus A/B/C, RSV A and B, bocaviruses (types 1 through 4), human metapneumovirus, coronavirus (229E, NL63, and OC43), and enterovirus, thereby offering a comprehensive diagnostic panorama [33].

Definition of exposed or Unexposed Group: In this study, we defined pediatric patients who received antimicrobial treatment for at least 48 h during their hospital stay as the “antibiotics-exposed” group. Conversely, those who did not meet this criterion were classified into the “unexposed” group.

Statistical Analysis: Statistical analyses were conducted using R software (version 4.1.1, 10 August 2021). Baseline demographic and clinical characteristics of patients were detailed using the ‘table one’ package, producing an informative “Table 1”. This table systematically presents categorical variables as counts and percentages, and continuous variables as means and standard deviations (for normal distributions) or medians and interquartile ranges (for non-normal distributions), depending on their distribution. Categorical data were analyzed using the Chi-square test or Fisher’s exact test when more appropriate, while continuous variables were analyzed using one-way ANOVA or the Kruskal–Wallis test, with the t-test or Wilcoxon test used for two-group comparisons. Statistical significance was determined at a *p*-value < 0.05 with a 95% confidence interval.

The study applied the ‘twang’ R package, which utilizes advanced machine learning algorithms, to derive propensity scores [34]. This method was used to balance baseline characteristics between antibiotic-exposed and non-exposed groups. The analysis used multivariate logistic regression with antibiotic factors (Abx^0^ and Abx^1^) as the outcomes and the baseline variables as potential confounders. The balance of covariates before and after matching was assessed using standardized mean differences, with a threshold of 0.15 indicating a satisfactory balance. Where this threshold was exceeded, the analysis recommended combining propensity score adjustments with further covariate adjustments, enhancing the reliability of the conclusions. In fact, the gradient boosting machine (GBM) was employed exclusively for the purpose of propensity score matching to adjust for confounding variables, rather than for predictive modeling.

Secondary outcomes, such as the time until fever resolution, were analyzed using the ‘survival’ and ‘survminer’ R packages, with the log-rank test assessing differences. This robust statistical approach helped clarify the impact of antimicrobial therapy on the speed of recovery and reduction of hospital stays in non-critical pediatric patients with acute viral respiratory tract infections.

## 3. Results

Patient Demographics and Clinical Characteristics: Our cohort included 238 patients with a median age of 15 months (IQR, 7 to 37 months) and a slight female predominance (52.9%). The median weight was 10.00 kg (IQR, 6.91 to 14.07 kg). Cough was the most common symptom (74.2%), followed by shortness of breath (68.5%), and fever (52.5%). The median illness duration before presentation was 3 days (IQR, 2 to 4 days). The median maximum temperature was 37.95 °C (IQR, 37.23 to 38.90 °C). Vomiting and diarrhea were reported in 37.8% and 11.8% of patients, respectively, while convulsions were less common at 6.3%. Comorbid conditions were present in 34.3% of patients, and co-infections were seen in 26.9%. (Table 1).

Laboratory and Radiological Findings: The median white blood cell count was 11.30 × 10^9^/L (IQR, 8.70 to 15.80 × 10^9^/L). The median hemoglobin level was 12.00 g/dL (IQR, 10.50 to 12.80 g/dL), and the median platelet count was 330.5 × 10^9^/L (IQR, 271 to 430 ×10^9^/L). C-reactive protein had a median value of 14.8 mg/L (IQR, 5.53 to 39.8 mg/L). Median values for albumin, creatinine, and alanine aminotransferase were 36 g/L (IQR, 32 to 39 g/L), 40 µmol/L (IQR, 36 to 46 µmol/L), and 26 U/L (IQR, 18 to 35.5 U/L), respectively. Radiologically, 54.6% of the X-rays were abnormal. (Table 1).

Therapeutic Interventions and Length of Hospital Stay: Antibiotics were used in 57.6% of patients, while 28.2% required respiratory support. The median hospital stay was 4 days (IQR, 3 to 7 days) (Table 1).

Viral Infections and Co-Infections: Among the 238 pediatric patients, viral infections were PCR-confirmed from nasopharyngeal aspirates. Human rhinovirus/enterovirus (HRV/EV) was the most prevalent, present in 44.5% of the cohort, followed by respiratory syncytial virus (RSV) in 18.1% of cases. Co-infections showed HRV/EV and RSV together in 8.4% of patients, while HRV/EV and parainfluenza virus co-occurred in 5.0% of cases. Other viruses such as influenza A Virus, parainfluenza virus, adenovirus, and human metapneumovirus were present in smaller proportions. (Figure 1).

Propensity Score Weighting Analysis for causal impact of antibiotics on length of stay: To address potential imbalances between the two groups (Abx^0^ and Abx^1^), we employed propensity score weighting using the ‘twang’ R package, which leverages a generalized boosting machine algorithm. This technique allowed us to account for observed covariates and approximate randomized control trials in observational data. Before weighting, we identified significant imbalances in several covariates. Covariates with an SMD of 0.15 or less were considered balanced between the two groups, while those with an SMD greater than 0.15 were deemed imbalanced. Balanced covariates included duration of illness (DOI), shortness of breath (SOB), vomiting, convulsion, neutrophil count, platelet count, creatinine level, ALT level, and requirement of respiratory support. Imbalanced covariates included age, weight, gender, fever, cough, maximum temperature, wheeze, diarrhea, co-morbidity, co-infection, WBC, hemoglobin, lymphocyte count, CRP level, albumin level, X-ray result, and length of stay (LOS). To correct for these imbalances, we applied propensity score weighting to estimate the average treatment effect on the treated (ATT) (Table 2).

Summary statistics before and after GBM model: After applying the generalized boosted model (GBM) for propensity score matching, we ensured better balance between the treated (137 patients) and control (101 patients) groups. Initially, significant imbalance was indicated by a maximum effect size of 0.612. After matching, the maximum effect size decreased to 0.461, and the mean effect size reduced from 0.220 to 0.122, with improvements in the Kolmogorov–Smirnov statistic as well. This indicates that the matching process significantly improved group comparability, reducing potential biases in our analysis. The model ran 4285 cycles to adjust and improve the matching between the groups, ensuring that the treated and control groups were as similar as possible in terms of baseline characteristics. This extensive process ensured the accuracy and reliability of our results (Table 3).

Propensity score weighting data with visualization: We enhanced the comparison between the Abx0 and Abx1 groups, achieving a balance across 12 factors, each with small differences (standardized mean differences or SMDs of 0.15 or less) using a method called propensity score weighting. Despite this, residual imbalances remained for fever, diarrhea, C-reactive protein, and albumin, suggesting that these factors might have influenced the outcomes and warranted careful interpretation. This analysis underscored the efficacy of propensity score weighting in enhancing the validity of causal inferences in observational studies, as illustrated in the Love plot, which elegantly encapsulated the SMD of 16 covariates before and after matching. (Table 4, Figure 2).

Bivariate Analysis: In our bivariate analysis comparing Abx0 and Abx1 cohorts, significant differences were observed in age, white blood cell count, and length of hospital stay, each validated by a *p*-value < 0.05. Neutrophil count did not show a significant difference (Figure 3).

Length of Hospital Stay Analysis: Using a survey-weighted linear regression model, we found that antimicrobial administration corresponded to a longer length of stay, approximately 2.19 days. Diarrhea was inversely correlated with the length of stay, reducing it by roughly 2.26 days, while higher albumin levels were associated with shorter stays. No significant relationships were observed between length of stay and either fever or C-reactive protein (Table 5).

Time to Recover from Fever: A Kaplan–Meier analysis comparing the time to recover from fever between the Abx0 and Abx1 groups showed no significant difference (*p*-value = 0.37). A log-ranked test suggested the hazard ratio was 0.84 (CI: 0.57–1.2), indicating that antibiotic use did not significantly impact the time to recover from fever (Figure 4 and Figure 5).

## 4. Discussion

HRV/EV and RSV Infection Prevalence: Our research focused on the prevalence of respiratory viruses in Saudi Arabian children. We found that 44.5% of non-critical cases tested positive for human rhinoviruses/enteroviruses (HRV/EV). This figure is on the higher end of reported prevalence rates, which have varied due to differences in detection methods, sample sizes, or geographic influences. For example, Launes et al. reported a prevalence of 57.3% [53], while Jacobs et al. [54] and Comte et al. [55] found lower rates of 33% and 28.4%, respectively. Studies by Neumann et al. [56], Bouvet et al. [57], and Duclos et al. [58] reported even lower prevalence rates, likely due to demographic, seasonal, or geographical factors. Respiratory syncytial virus (RSV) is another major cause of pediatric hospitalizations [59]. It accounts for 10% to 28% of infant hospitalizations. Different studies have reported varying prevalence rates, with Reeves et al. [60] and Hacımustafaoğlu et al. [61] finding RSV in approximately 26% of children.

The Effects of Co-Infections: We observed that HRV/EV, RSV, and PIV often co-infect, potentially impacting the course and prognosis of diseases. Our findings, consistent with earlier studies, show that co-infections are common [62,63,64]. In children under five, significant pathogens include RSV, HMPV, and PIV. However, unlike other studies, we did not find any cases of HMPV-RSV co-infections [65]. Hospital stays and fatality rates were similar across single viral infections and viral co-infections [66]. Our co-infection rate of 8.4% was lower than those reported by other studies [67,68].

Hospital Stay and Antibiotic Use: Our analysis of pediatric patients with acute viral respiratory tract infections (AVRTI) revealed that antibiotic use increased the length of stay (LOS) by an average of 2.19 days. This finding aligns with previous research indicating that identifying the virus does not always lead to effective treatment and is often correlated with abnormal X-ray results [69]. In contrast, studies suggest that a PCR multiplex respiratory panel can reduce LOS and unnecessary antibiotic use by quickly identifying viral causes [34,41]. Careful antibiotic use is crucial, as most children with viral RTIs recover without antibiotics, avoiding complications and resistance [39,40]. Although a PCR multiplex respiratory panel can improve diagnosis speed, it may not significantly impact LOS or antibiotic use, although it is considered cost-effective and beneficial for patient outcomes [42,70].

Impact of Antibiotics on Fever Recovery: Our study investigated the impact of antibiotics on the duration of fever recovery in children with acute viral respiratory tract infections. Kaplan–Meier curve analysis, confirmed by the log-rank test, showed no significant difference between antibiotic-treated and non-treated patients. Antibiotics are frequently prescribed for respiratory diseases, despite their limited effectiveness against these conditions [43,44]. Misconceptions, such as the belief that antibiotics shorten the duration and lessen the severity of viral infections, contribute to this practice [47]. In both the United States and Canada, a significant rate of antibiotic misuse for viral upper respiratory infections (URIs) has been documented, ranging from 40% to 75%, often driven by the presence of fever above 38.5 °C [48]. In Saudi Arabia, 27% of parents sought antibiotics for their children during fever episodes [23]. Antibiotics are typically not recommended for viral URIs [62]. Studies have shown no significant difference in clinical outcomes [71].

Potential Clinical Consequences: Our research highlights important changes in the treatment of pediatric respiratory infections. We urge cautious antibiotic prescribing, especially for viral infections, to reduce the risk of antibiotic resistance and unwanted side effects. Given the correlation between antibiotic use and extended hospital stays, a re-evaluation of admission criteria is warranted. The efficacy of the PCR multiplex respiratory panel suggests ways to improve diagnostic methods, while the poor effect of antibiotics on fever recovery calls for a re-evaluation of current fever management practices.

Strengths: Our research combined clinical investigations with advanced machine learning methods, specifically the gradient boosting machine algorithm. Our primary aim was to address antibiotic resistance, employing state-of-the-art analytical tools and robust statistical procedures, including Kaplan–Meier curves and the log-rank test. Our study explored antibiotic stewardship, multiple viruses, and co-infections, with fever as a key symptom.

Weaknesses: The technical terms in our study’s title might limit its appeal to a broader audience. Retrospective cohort studies, while effective for identifying patterns, have limitations such as recall bias and incomplete information. The small sample size reduces statistical power, and reliance on electronic medical records may introduce inaccuracies. Data collection from a single medical facility further limits generalizability. Additionally, the inability to distinguish between upper and lower respiratory tract infections affects the granularity of the analysis. Future research can address these issues by improving data collection methods, increasing sample sizes, and considering a broader spectrum of pathogens.

## 5. Conclusions

Our findings highlight the need for caution when prescribing antibiotics due to their potential harmful effects and the limitations of some diagnostic procedures. Given that antibiotics are often ineffective against viral illnesses, fever alone should not warrant antibiotic treatment. Our research underscores the importance of evidence-based antibiotic stewardship to combat the growing problem of antibiotic resistance. Targeted and prudent use of antibiotics can improve clinical outcomes for young patients and address a significant global health risk.

## Figures and Tables

**Figure 1 antibiotics-13-00518-f001:**
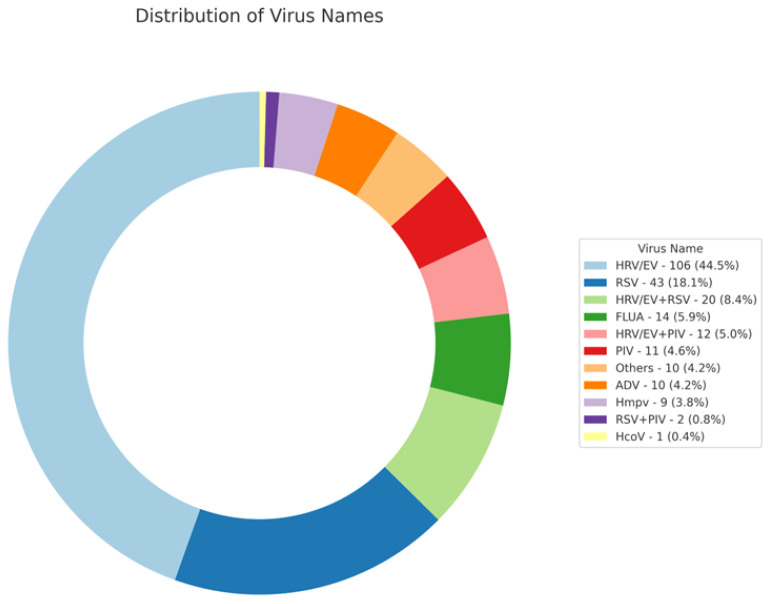
Detected virus from a PCR multiplex respiratory panel. HRV/EV = human rhinovirus/enterovirus RSV = respiratory syncytial virus, FLUA = influenza A, Hmpv = human metapneumovirus, parainfluenza virus = PIV, ADV = adenovirus, Hmpv = human metapneumovirus, HcoV = human coronavirus.

**Figure 2 antibiotics-13-00518-f002:**
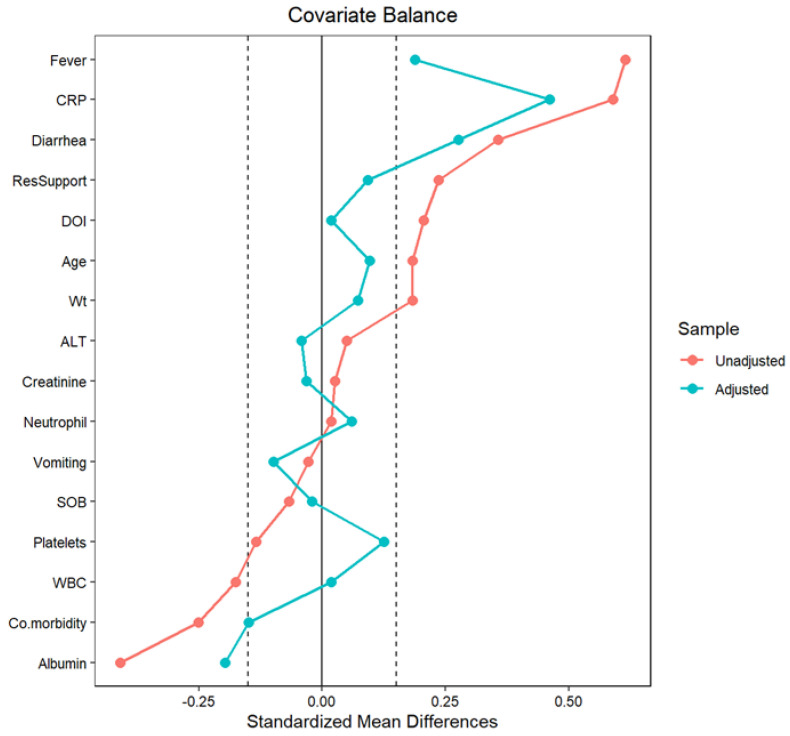
Love plot to demonstrate the standardized mean difference of sixteen covariates before and after propensity score matching. The red line represents the unadjusted, and the blue line adjusted covariates. Both shade lines are the threshold cut points for covariates balance standardized mean difference (−0.15 to +0.15).

**Figure 3 antibiotics-13-00518-f003:**
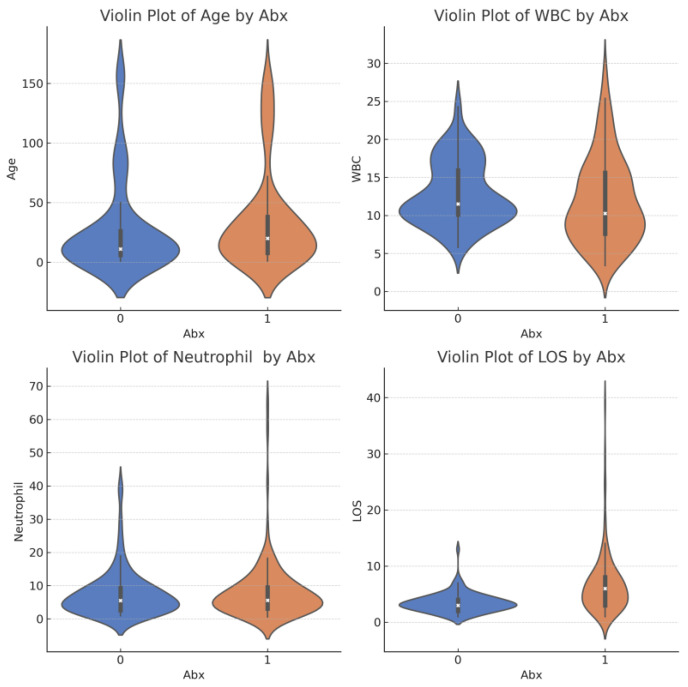
Violin plot comparing the distributions of age, WBC (white blood cells), neutrophil, and length of stay (LOS) between groups with and without antibiotic treatment (Abx, where 1 = Yes and 0 = No). Each subplot shows the spread and density of values for these variables, helping to visualize any differences based on antibiotic use.

**Figure 4 antibiotics-13-00518-f004:**
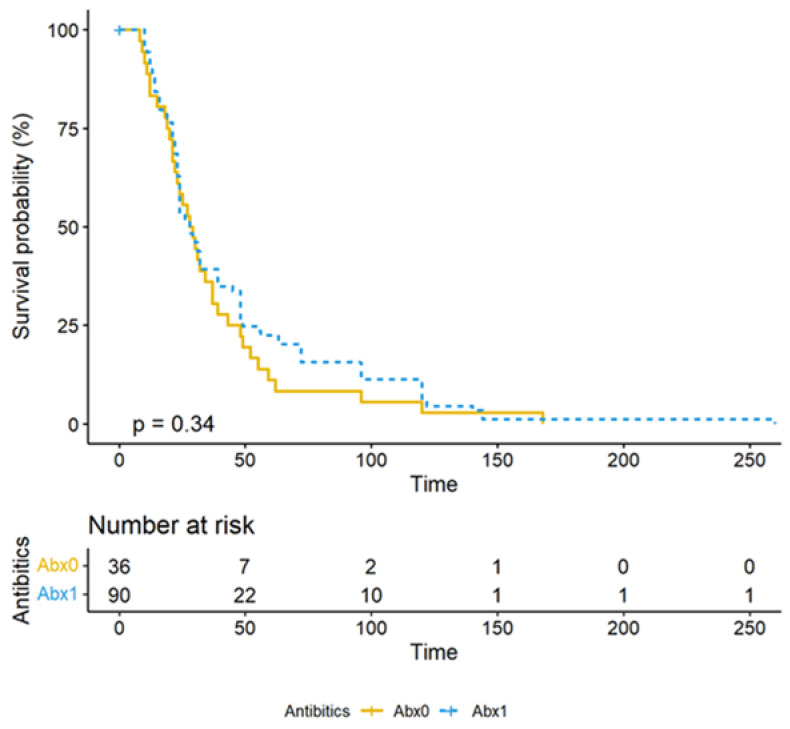
The Kaplan–Meier curve depicting time to recovery from fever. It shows the comparison between two groups: Abx^0^ (those who did not receive antibiotics) and Abx^1^ (those who did). The inferential interpretation of this curve revealed there was no significant difference between the two groups, with a *p*-value of 0.34. So, whether antibiotics were used, the time to recovery from fever was similar.

**Figure 5 antibiotics-13-00518-f005:**
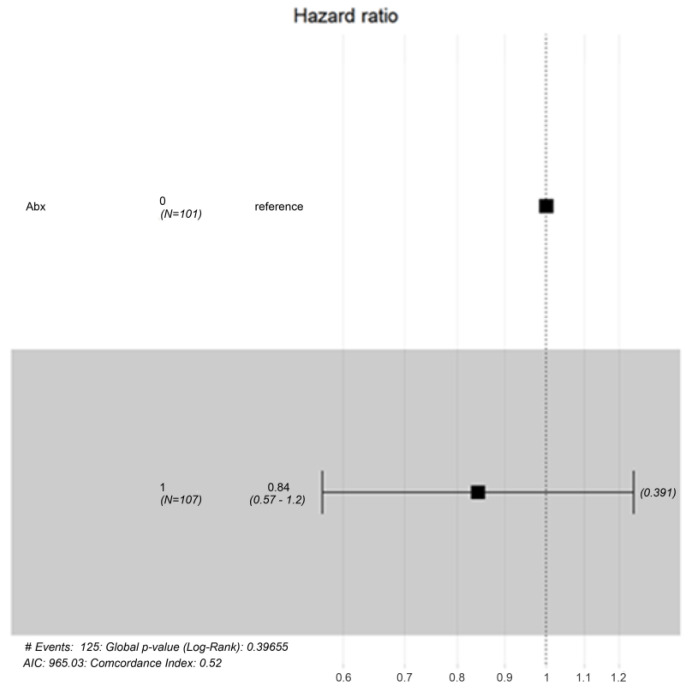
Log-ranked test showing comparison between the Abx1 antibiotics groups and non-antibiotics groups (Abx0), recovery time of fever.

**Table 1 antibiotics-13-00518-t001:** Demographic, clinical, and laboratory data for 238 patients. Age in months (median, IQR); gender as % male. Clinical symptoms (fever, cough, SOB, wheezing, vomiting, diarrhea, convulsions) and interventions (antibiotics, respiratory support) are reported as percentages. Median and IQR are used to present weight, days of illness, maximum temperature, and length of stay. Laboratory values include WBC, Hb, neutrophils, lymphocytes, platelets, CRP, albumin, creatinine, and ALT (median, IQR). X-ray results are shown as frequency and percentage.

Total Number (N)		N = 238
Age (median [IQR])		15.00 [7.00, 37.00]
Gender, n (%)	Male	112 (47.1)
Wt (median [IQR])		10.00 [6.91, 14.07]
DOI (median [IQR])		3.00 [2.00, 4.00]
Fever, n (%)	Yes	125 (52.5)
MaxTem (median [IQR])		37.95 [37.23, 38.90]
Cough, n (%)	Yes	175 (74.2)
SOB, n (%)	Yes	163 (68.5)
Wheeze, n (%)	Yes	122 (51.7)
Vomiting, n(%)	Yes	90 (37.8)
Diarrhea, n(%)	Yes	28 (11.8)
Convulsion, n(%)	Yes	15 (6.3)
Comorbidity, n (%)	Yes	80 (34.3)
Co-infection, n (%)	Yes	63 (26.9)
WBC (median [IQR])		11.30 [8.70, 15.80]
Hb (median [IQR])		12.00 [10.50, 12.80]
Neutrophil (median [IQR])		5.58 [3.10, 9.32]
Lymphocyte (median [IQR])		3.90 [2.18, 6.08]
Platelets (median [IQR])		330.5 [271, 430]
CRP (median [IQR])		14.8 [5.53, 39.8]
Albumin (median [IQR])		36 [32, 39]
X-Ray result, n (%)	Abnormal	130 (54.6)
Creatinine (median [IQR])		40 [36, 46]
ALT (median [IQR])		26 [18, 35.5]
Abx, n(%)	Yes	137 (57.6)
Res support, n (%)	Yes	67 (28.2)
LOS (median [IQR])		4.00 [3.00, 7.00]

**Table 2 antibiotics-13-00518-t002:** Characteristics of covariates used in the generalized boosted model before matching.

Variable	Abx^0^	Abx^1^	SMD
N	101	137	
Age (median [IQR])	11 [6, 26]	20 [7.75, 38.00]	0.183
Gender (%), Male	35 (34.7)	77 (56.2)	0.443
Wt (median [IQR])	9.45 [6.34, 13.25]	10.95 [7.73, 15.62]	0.174
Duration of illness (DOI) (median [IQR])	3.0 [1.0, 3.0]	3.0 [2.00, 5.0]	0.135
Fever (5), Yes	36 (35.6)	89 (65.0)	0.613
Maximum temp (MaxTem) (median [IQR])	37.5 [37.1, 38.3]	38.4 [37.6, 39.1]	0.634
Cough (5), Yes	70 (70.0)	105 (77.2)	0.164
Shortness of breath (SOB) (%), Yes	71 (70.3)	92 (67.2)	0.068
Wheeze (%), Yes	57 (57.0)	65 (47.8)	0.185
Vomiting (%), Yes	39 (38.6)	51 (37.2)	0.029
Diarrhea (%), Yes	4 (4.0)	24 (17.5)	0.449
Convulsion (%), No	101 (100.0)	137 (100.0)	<0.001
Co-morbidity (%), Yes	40 (40)	40 (30.1)	0.209
Co-infection (%), Yes	39 (39.4)	24 (17.8)	0.493
WBC (median [IQR])	11.50 [10.10, 15.90]	10.25 [7.57, 15.60]	0.189
Hb (median [IQR])	12.43 [11.70, 12.80]	11.30 [10.17, 12.53]	0.552
Neutrophil (median [IQR])	5.52 [2.61, 9.30]	5.60 [3.11, 9.45]	0.062
Lymphocyte (median [IQR])	4.94 [3.22, 8.01]	3.50 [2.07, 5.45]	0.361
Platelets (median [IQR])	332 [270, 490.25]	330.5 [271, 388.5]	0.140
CRP (median [IQR])	8 [2.6, 25.8]	27.7 [10.1, 68.1]	0.836
Albumin (median [IQR])	37 [33, 39]	35 [30, 39]	0.481
X-Ray result (%) Yes	40 (39.6)	90 (65.7)	0.541
Creatinine (median [IQR])	39.5 [36, 46]	40.5 [36, 45.3]	0.015
ALT (median [IQR])	25.00 [19.00, 34.00]	26.00 [18.00, 37.00]	0.024
ResSupport (%)	101 (100.0)	137 (100.0)	<0.001
Length of stay (LOS) (median [IQR])	3.00 [2.00, 4.00]	6.00 [3.00, 8.00]	0.793

**Table 3 antibiotics-13-00518-t003:** Summary statistics before and after GBM model.

	n.treat	n.control	ess. treat	ess. control	max.es	mean.es	max. ks
unw	137	101	137	101	0.612	0.220	0.340
es. mean.ATT	137	101	137	41	0.461	0.122	0.240
	mean. ks		iteration				
unw	0.164		NA				
es. mean.ATT	0.110		4285				

Unw= unweighted, n. treat= number of treats, n. control= number of controls, ess. treat= effective sample size of treated, ess. Control= effective sample size in control, max.es= maximum effect size, mean.es= mean effect size, KS= Kolmogorov Smirnov statistics, ATT= average treatment effect on treated.

**Table 4 antibiotics-13-00518-t004:** Balance measures after using the twang R package (Toolkit for Weighting and Analysis of Non-equivalent Groups) where a generalized boosted machine algorithm was used to obtain the cuff up the balance of SMD of 0.15 among the covariates.

	Type	Diff.Adj	M.Threshold
Propensity score	Distance	4.2665	
Age	Continuous	0.0964	Balanced, <0.15
Weight	Continuous	0.0731	Balanced, <0.15
DOI	Continuous	0.0184	Balanced, <0.15
Fever	Binary	0.1885	Not Balanced, >0.15
SOB	Binary	−0.0207	Balanced, <0.15
Vomiting	Binary	−0.0985	Balanced, <0.15
Diarrhea	Binary	0.2761	Not Balanced, >0.15
Co-morbidity	Binary	−0.1490	Balanced, <0.15
WBC	Continuous	0.0189	Balanced, <0.15
Neutrophil	Continuous	0.0601	Balanced, <0.15
Platelets	Continuous	0.1250	Balanced, <0.15
CRP	Continuous	0.4608	Not Balanced, >0.15
Albumin	Continuous	−0.1972	Not Balanced, >0.15
ALT	Continuous	−0.0414	Balanced, <0.15
Creatinine	Continuous	−0.0313	Balanced, <0.15
ResSupport	Binary	0.0932	Balanced, <0.15
Balance tally for standardized mean differences			
Balanced, <0.15	12		
Not Balanced, >0.15	4		

**Table 5 antibiotics-13-00518-t005:** Survey-weighted linear regression output. Fever, diarrhea, CRP, and albumin were included in the model because of SMD > 0.15 for minimizing bias.

Model Information				
Observation 238				
Dependent Variable: Length of Hospital Stay (LOHS)				
Type: Survey-weighted linear regression				
MODEL FIT: R^2^ = 0.37				
Adj. R^2^ = 0.36				
Standard errors: Robust				
	Est	S. E	t val	*p* value
(Intercept)	1.86	3.01	6.19	0.00
Abx	2.19	0.53	4.11	0.00
Fever	−0.37	0.63	−0.59	0.56
Diarrhea	−2.26	0.55	−4.08	0.00
CRP	−0.00	0.00	−1.06	0.29
Albumin	−0.40	0.08	−4.90	0.00
Estimated dispersion parameter = 15.58				

Abx = Antibiotics.

## Data Availability

Data will be supplied on reasonable request.

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
