# Peer review of "Exploring the Impact of Antibiotics on Fever Recovery Time and Hospital Stays in Children with Viral Infections: Insights from Advanced Data Analysis"

_antibiotics, 2024, doi:10.3390/antibiotics13060518_

Round 1
Reviewer 1 Report
Comments and Suggestions for Authors
Dear Mohammed Qahtani and Colleagues,
Your aims to show the association of overuse of antibiotics for viral respiratory tract infections and clinical outcomes for children presenting to a single centre. Below are several areas that should be addressed to strengthen your paper.
1) Over a 5 year period, 238 patients were included. Can you provide more detail as this seems low given the number of paediatric admissions annually? There seems to be a risk of selection bias.
2) The methods outline a number of clinical parameters and demographic data that were collected but this is not presented in the results. There is essentially no description of the population studied here. For example, what is the average age? CXR findings? Proportion of bacterial co-infections? Table 1 is confusing because this is not a table of demographics and I'm not sure what these values are. Are these the linear predictors for each co-variate in the model? If so, that is an odd thing to present and requires explanation to help the reader.
3) Why is a machine learning technique used rather than predictive modelling such as regression analysis? This is not a large dataset.
4) What measures were taken to avoid overfitting of the model and corrections for multiple testing?
5) The results section is very confusing to read as the authors repeatedly quote multiple outputs from the GBM model rather than summarising the results and providing an interpretation that helps the reader. For example the reporting of Table 2 gives a description of the GBM output but does not help the reader understand what this actually means. This is the case for all of the results derived from the GBM model. Why are numbers repeatedly reported to 7 decimal places?
6) Confounders include laboratory markers. Were other confounders such as co-morbidities considered?
7) Figure 4 - what do the different colours mean?
8) The discussion is long.
Comments on the Quality of English LanguageThe English language writing is very good but I would encourage the authors to tone down some of the language. The words 'meticulous' for example is used a lot!
Author Response
Dear Reviewer,
Thank you for your insightful comments and the opportunity to clarify the key aspects of our study titled "Exploring the Impact of Antibiotics on Fever Recovery Time and Hospital Stays in Children with Viral Infections: Insights from Advanced Data Analysis."

Reviewer 2 Report
Comments and Suggestions for Authors
Dear authors. I received the entitled: Exploring the Impact of Antibiotics on Fever Recovery Time and Hospital Stays in Children with Viral Infections: Insights from Advanced Data Analysis for review. Despite the relevance, some points need to be reviewed.
1. Abstract- Please describe setting of study ( pediatric wards)
2. Introduction is too generic- Could be improved , describing more studies about length of stay and fever recovery. I also didn't find the article's aim
3. Material and methods- This section needs to changed in order to appear before the results and not after.
a) Please explain better why did you choose the age group from one month to 14 years
b) Please explain better the dynamics of children care within the wards. Do children need to wait for PCR results to receive antibiotics, or empirical antibiotics are prescribed until PCR results?
c) Acute viral respiratory tract infection is not well defined. Please review it in order to define which are considered as VRTI
d) Definition of antibiotic-exposed group and unexposed should be improved. It's not clear for the reader if when antibiotic prescription were included. In the first days of admission, or in any moment of admission?
4. Results
All tables should be reviewed and include legends for CRP, ALT, DOI, SOB and etc.
5. Conclusion-
It's too long, could be shortened in order to answer the aim of the article.
Comments on the Quality of English LanguageMinor editing are necessary
Author Response

(The authors gave the same response as above.)

Round 2
Reviewer 1 Report
Comments and Suggestions for Authors
Dear Dr Al Qahtani,
Thanks for the substantial revisions that you have made to the manuscript. It is much easier to read now.
I'd be grateful if you would consider some further revision.
1) In your response you state that the GBM is solely for the purpose of propensity matching. I cannot comment on the merits of that approach versus a more tradition approach. However, I think there are elements of the manuscript that are misleading in this regard - ie. you have not carried out an advanced ML model to predict outcomes etc. I think this needs to made clear in paragraph 3 of 'materials and methods' to explain why you used GBM and what you used it for. I think 'advanced data analysis' in the title is an overstatement.
2) Thank you for including more demographic information. Table 2 is much more meaningful now that you report summary statistics for these characteristics rather than outputs from the GBM output. However there is now overlap between table 1 and 2. suggest Table 1 is unnecessary.
The first two paragraphs of results describing the demographics is unnecessarily long. you can refer the reader to table 2.
3) Table 3 and its associated text is still confusing. Does this need to be in the text or can it be moved to a supplementary section? All the reader needs to know is whether the groups are matched or not. And you cannot expect the reader to have any knowledge of GBM to work this out. You must interpret this output for the reader.
4) Length of hospital stay and time to recovery analysis are you key results. But they are buried at the end of the results section. the earlier parts need to be thinned out to make this section stand out more.
5) regarding weakness of your study. If you identify your study's title as a weakness at this stage, then I suggest you change it (or remove that comment).
6) Suggest revising presentation in tables. 'Yes (%)' to 'n(%)'
Author Response
Dear,
Thank you for the opportunity to revise our manuscript titled "Exploring the Impact of Antibiotics on Fever Recovery Time and Hospital Stays in Children with Viral Infections: Insights from Advanced Data Analysis" and for the insightful comments and suggestions provided by you. We have carefully considered each point raised and have made corresponding revisions to the manuscript.
With thanks
Corresponding Author

Reviewer 2 Report
Comments and Suggestions for Authors
Almost all queries were improved.
There's still necessity to improve information about studies related to length of stay and fever recovery in introduction section .
I have serious concerns about using clinical judgment and a broad spectrum respiratory symptoms to classify children as acute viral respiratory tract infection. In fact, in a real-world situation, almost all the healthcare services use clinical judgment, but the challenge is how to discharge completely coinfection with bacteria in these setting and conclude that antibiotics are unnecessary?
Comments on the Quality of English LanguageMinor edition are necessary
Author Response

(The authors gave the same response as above.)

Round 3
Reviewer 2 Report
Comments and Suggestions for Authors
Dear authors. Information about descrpition or more studies about length of stay and fever recovery in introduction section was not answered.
Comments on the Quality of English LanguageNone to declare
Author Response
Dear Reviewer,
We greatly appreciate your thorough review and valuable feedback on our manuscript. We have carefully considered your suggestions and made the necessary revisions to the introduction section to enhance clarity and conciseness while maintaining the depth and detail of the original content.
With thanks.
Corresponding Author.
